# Effect of Aggression on Peer Acceptance Among Adolescents During School Transition and Non-Transition: Focusing on the Moderating Effects of Gender and Physical Education Activities

**DOI:** 10.3390/ijerph16173190

**Published:** 2019-09-01

**Authors:** Myoungjin Shin, Changhyun Lee, Yongsik Lee

**Affiliations:** 1Research Institute for Basic Sciences, Soonchunhyang University, Eumae-ri, Sinchang-myeon, Asan-si 315358, Korea; 2Department of Physical Education, Catholic Kwandong University, 24, Beomil-ro 579beon-gil, Gangneung-si 25601, Korea; 3Faculty of Sport and Leisure Studies, Catholic Kwandong University, 24, Beomil-ro 579beon-gil, Gangneung-si 25601, Korea

**Keywords:** physical education activities, peer acceptance, aggression, school transition, adolescents

## Abstract

The present study examined the effect of aggression on peer acceptance among adolescents. We focused on the moderating effects of gender and participation in physical education activities and examined whether these effects varied during school transition. We used longitudinal data of adolescents aged 10 to 17 years that were obtained from a survey that was conducted by the National Youth Policy Institute. In only early adolescence, the interaction effect of gender and physical education activity influenced the relationship between aggression and peer acceptance. Specifically, the negative relationship between aggression and peer acceptance was strengthened among female students who participated in physical education activities as compared to female students who did not. This effect was not observed in male students. However, during transition from primary to secondary school, the negative effect of physical education activities did not exist. For middle-adolescents, for whom physical education activities increased more than previous years, the negative relationship between aggression and peer acceptance worsened. These influences were the same, regardless of gender. Thus, this study suggests that physical education activities improve the negative relationship between aggression and peer acceptance during school transition.

## 1. Introduction

Peers greatly influence positive social development, as peer interaction gradually increases in adolescence in comparison to childhood [1,2]. Of the various behavior patterns, aggression is an important factor that affects positive social development. Students who show high aggression in class face low acceptance from peers [3,4,5], and low peer acceptance negatively affects the students’ psychological and social development and school adjustment [6,7,8]. In this study, we examined whether physical education helps to improve the negative effect of aggression on peer acceptance for students in their early teens and whether this effect is identical for both genders. We also examined whether physical education activities can improve the negative relationship between aggression and peer acceptance during school transition, where students interact with new people more than during non-transition.

### 1.1. Role of Gender in the Association between Aggression and Peer Acceptance

Peer acceptance is an important aspect of peer relationships and it can be divided into peer status and popularity [8]. Peer status refers to one being recognized as a valuable member among friends in terms of leadership or status, while popularity refers to one being liked by or getting along with friends. In contrast, friendship, which is another aspect of peer relationships, is a dyadic relationship that requires voluntary participation [9], and it can explained by the existence or lack of close friends, the number of close friends, or the quality of the relationship. Peer acceptance and friendship are interrelated concepts that explain peer relationship and they are not necessarily the same [10,11]. In other words, this indicates that, if one is preferred by the peer group, it does not necessarily mean that they have a satisfactory friendship; even those who are not well-accepted by the peer group can have a close friendship.

Aggression is a negative behavior or stimulation that displays the intention and anticipation of causing harm to another person [12,13]. Although the definitions of aggression types vary for each scholar, aggression can be categorized into overt aggression and relational aggression. Overt aggression refers to physically harming or threatening others through actions, such as hitting, pushing, or threatening. In contrast, relational aggression refers to impairing peer relationship and harming the perception of inclusion by using behaviors, such as spreading rumors, ignoring or not talking, or ostracizing from the group [14].

Aggression has a strong relationship with peer acceptance, and high aggression is known to predict low peer acceptance [3,4,5,15]. In middle childhood, = overt and relational aggression are both related to low peer acceptance [16,17,18].

Although, generally, aggression predicts low peer acceptance in children or teenagers [19], the form of aggression varies according to gender. In many studies, female students showed more relational aggression than male students [14,20,21], and the effect of relational aggression on peer acceptance was greater among the female than male students [15]. In the study by LaFontana and Cillessen [22], the relationship between relational aggression and social preference was not significant in male students, while it had a statistically significant negative relationship for female students. Puckett, Aikins, and Cillessen [23] also showed that the tendency for relational aggression to reduce social preference was stronger among female students.

Overt aggression is predicted to have a small difference between genders, while the negative impact of relational aggression on peer acceptance was stronger for female students. Henington, Hughes, Cavell, and Thompson [24] showed that female students with high overt aggression were rejected by peers, regardless of the level of relational aggression in early childhood; the same results were observed for male students with overt and relational aggression. A study on the effect of overt aggression on peer rejection in early adolescents showed no difference between genders [3]. Although aggression, in general, has a negative impact on peer acceptance in childhood and adolescence, the degree to which overt aggression predicts peer acceptance does not differ by gender, unlike relational aggression. Therefore, we must consider differences by gender in order to gain a clear insight into the relationship between aggression and peer acceptance.

### 1.2. Influence of Physical Education Activities on Peer Relationships

Identifying the moderating variable that improves the negative relationship between aggression and peer acceptance is important in preventing peers from inducing harmful outcomes (example: maladaptation to school, psychological/social negative impact). In this study, the main moderating variable is a physical education activity or sports participation. Physical education activities and sports participation have a positive influence on peer acceptance, and it can be explained by the social interdependence theory [25,26]. According to the social interdependence theory, cooperative conditions (where a common goal is achieved by members cooperating with each other) and competitive conditions (where members work together, but only one individual can achieve the goal) both result in the formation of positive social relationships. In a physical education activity or sports activity, a common goal must be achieved with other members rather than individually achieving the goal. In the process of achieving this goal, cooperative and competitive conditions are naturally formed, which thereby enables the formation of positive peer acceptance via a physical education activity and sports participation.

Many empirical studies have produced consistent findings that a physical education activity or sports participation has positive effects on a child’s peer acceptance. Sports activity participation among adolescents is highly related to positive peer acceptance [27], and children and teenagers who did not participate in sports had difficulty in forming social competence and they had lower self-control ability and self-confidence when compared to children who participate in sports [28,29]. In a study by Kennedy [30], the variable that had the largest effect on popularity was athletic status, and there was a strong perception that students with lower athletic ability are socially isolated [31]. Therefore, participation in a physical education activity or sports activity can negatively influence the relationship between aggression and peer acceptance. However, these positive effects are more likely to be observed in male students. Buchanan, Blankenbaker, and Cotton [32] asked students in grades 4 to 6 what was most important to be popular among friends. While the male students cited being good at sports, the female students cited having good school grades. Chase and Dummer [33] also showed different preferences for popularity by gender: male students placed the greatest importance on sports, while female students ranked the importance in the order of good looks, good at sports, and good grades. Additionally, the male students had a higher frequency of participating in physical competitive activities in comparison to female students [34]. Similarly, Fredricks and Eccles [35] showed that, while sports participation helped to improve the sociality of male students, this was not observed in female students.

### 1.3. The Development of Aggression and Peer Acceptance

Physical aggression peaks in early childhood and then gradually reduces; they get rid of their anger more via verbal aggression, which thus results in reduced physical aggression [19]. Relational aggression increased as social-cognitive skills developed (see [17,36]). The reason can be that, as children develop, physical aggression becomes less normative in the peer group and decreases, while relational aggression becomes normative and increases [37].

As children develop, their aggression patterns also vary; peer rejection was comparatively lower when aggression decreased, depending on the developmental timing. Peer rejection was comparatively stable, even after the transition from primary school to secondary school [38], and there was the probability of undergoing the same experience of peer rejection or preference after adolescence [39]. This indicates that the social relationship within peer groups remain the same, even when a student transitions from primary school to secondary school or from secondary school to high school, even when peer acceptance is low and peer rejection is high.

The sample must comprise early teens in primary, secondary, and high schools to examine the effect of aggression on peer acceptance. Additionally, the possibility of using methods that help adolescents to improve behaviors and social skills (example: physical education activity, sports participation) are high during school transition when renegotiation of social status is necessary [3]. Therefore, it can be expected that physical education activities will have the greatest positive impact on the negative relationship between aggression and peer acceptance among early teens during school transition.

### 1.4. Summary and Research Hypotheses

Adolescents with high aggressive disposition are predicted to have low peer acceptance when combining the results of previous research. Physical education activity can improve peer acceptance, and this effect is more likely to be observed in male students than in female students. Considerations regarding two developmental times should be considered to test the moderating effect of physical education activity in the relationship between aggression and peer acceptance. First, changes due to age or school level (primary, secondary, high school) should be examined together. Physical aggression tended to decrease as school age increases [19], although peer acceptance does not show a big difference in the transition from early to late adolescence [38,39]. Therefore, it is necessary to comprehensively examine the effect physical education activities have on improving the relationship between aggression and peer acceptance during this transition. Second, considerations must be made based on whether it is time for school transition or not. As Perron-Gélinas et al. [3] reported that sports participation is likely to be valuable during school transition when the renegotiation of social status is needed; the effect of physical education activity is predicted to be the largest during the transition from primary to secondary school and from secondary to high school.

Although some studies have been conducted based on a longitudinal perspective to examine the social-behavioral characteristics and psychological changes of children and adolescents [3,27], studies on the relationship between aggression and peer acceptance in early teens and during school transitions are limited. Previous studies [3,27] have only examined the effect of sports participation (T1) on self-esteem (T2) or the effect of aggression (T1) on peer rejection (T2) in two-time measurements within early, mid, or late adolescence. In our study, we analyzed two school transitions (primary school to secondary school and secondary school to high school) and three non-transitions (grade 4 to 5 in primary school, grade 2 to 3 in secondary school, grade 1 to 2 in high school) to examine the effect of physical education activity on aggression and peer acceptance among early teens.

## 2. Materials and Methods

### 2.1. Research Subjects

The present study utilized longitudinal data from the Korean Children and Youth Panel Survey (KCYPS) that the National Youth Policy Institute (NYPI) conducted. The KCYPS is a seven-year longitudinal study that tracked Korean children from childhood to adolescence from 2010 to 2016 while using panel data to efficiently provide comprehensive data on elementary and middle school students. KCYPS conducted the study at three starting points (grade 1 in primary school, grade 4 in primary school, grade 1 in middle school) over seven times (2010 to 2016). In our study, we generated five cohorts from three-panel data. One cohort was generated from the grade 1 panel in primary school (cohort 1 or C1), two cohorts (C2, C3) were generated from the grade 4 panel in primary school, and two cohorts (C4, C5) were generated from the grade 1 panel in middle school. A total of five cohorts with two school transition periods (C2, C4) and non-transition periods (C1, C3, C5) were used in the final analyses.

C1 and C2 that were used in the analyses represent late childhood and early adolescence; C3 and C4 represent middle adolescence; and C5 represents late adolescence. Data that were collected at Time 1 (grade 4 in primary school, age: 10 years) and Time 2 (grade 5 in primary school, age: 11 years) were used for C1 (a total of 1987 students with 1020 males). Data collected at Time 1 (grade 6 in primary school, age: 12 years) and Time 2 (grade 1 in secondary school, age: 13 years) were used for C2 (a total of 1947 students with 1022 males). Data collected at Time 1 (grade 2 in secondary school, age: 14 years) and Time 2 (grade 3 in secondary school, age: 15 years) were used for C3 (a total of 1889 students with 995 males). Data that were collected at Time 1 (grade 3 in secondary school, age: 15 years) and Time 2 (grade 1 in high school, age: 16 years) were used for C4 (A total of 1869 students with 957 males). Finally, data that were collected at Time 1 (grade 1 in high school, age: 16 years) and Time 2 (grade 2 in high school, age: 17 years) were used for the analysis of C5 (a total of 1985 students with 1015 males).

### 2.2. Measurement

#### 2.2.1. Perceived Physical Education Activity (PPEA)

Physical education is required in all elementary, middle, and high schools in Korea. The duration of PPEA was measured by one item [40], “How many hours did you spend on your physical activity from physical education in the last week?” on a five-point Likert scale (1 = none at all, 2 = one hour, 3 = two hours, 4 = three hours, and 5 = over four hours). Therefore, the higher the score, the greater the duration of PPEA.

#### 2.2.2. Degree of Perceived Physical Education Activity (DPPEA)

DPPEA is the value of PPEA (T2) minus PPEA (T1). A positive DPPEA indicates that PPEA at T2 is higher than PPEA at T1, which signifies an increase in the physical education activity as compared to the previous year.

#### 2.2.3. Perceived Aggressive Disposition (PAD)

Perceived aggressive disposition was measured with a total of three items [41]: “When I am not allowed to do something I want, I argue or fight”, “I often fight about unimportant things”, and “Sometimes I am mad all day”. These items were measured on a four-point Likert scale (1 = not at all, 4 = very much). It was found that C1′s α = 0.73 (T1), C2′s α = 0.77 (T1), C3′s α = 0.74 (T1), C4′s α = 0.69 (T1), and C5′s α = 0.70 (T1).

#### 2.2.4. Perceived Peer Acceptance (PPA)

One item of the peer status construct, “My friends like to follow my word when we play or have a group activity”, and another item for the popularity construct, “I can socialize with my classmates well”, were used to measure perceived peer acceptance [40]. These were measured on a 4-point Likert scale (1 = not at all, 4 = very much). For C1, α = 0.61 (T1) and 68 (T2); for C2, α = 0.63 (T1) and 66 (T2); for C3, α = 0.66 (T1) and 70 (T2); for C4, α = 0.60 (T1) and α = 0.58 (T2); and, for C5, α = 0.57 (T1) and α = 0.57 (T2).

#### 2.2.5. Socioeconomic Status (SES)

We examined potential confounding effects of family socioeconomic status, which has been related to youngsters’ sports participation as well as to their social standing in the peer group [42,43]. To control for the confounding effect of SES in our study, we asked a single question, “Which of the following represents your home’s financial level?” This was measured on a seven-point Likert scale (1. Very low income, 2. Low income, 3. Slightly low income, 4. Average income, 5. Slightly high income, 6. High income, 7. Very high income). SES was measured at Time 1 for all five cohorts.

### 2.3. Data Analyses

A hierarchical multiple regression analysis was used to identify the influence that gender and physical education activity had on the effect of aggression on peer acceptance. In Step 1, various control variables (gender, SES, peer acceptance (T1), and PPEA (T1)) were included. In Step 2, we examined the effect of aggression on peer acceptance. Finally, in Step 3, we used two-way and three-way interaction variables to test the moderating effects of gender and physical education activity. Statistical significance level was set to 0.05. The analyzed data were found to meet four assumptions of multiple regression analysis: 1. linearity of the phenomenon measured, 2. constant variance of the error terms, 3. independence of the error terms, and 4. normality of the error term distribution. As shown in Table 1, Table 2, Table 3, Table 4 and Table 5, there is no high correlation between variables, so the multicollinearity problem does not exist.

## 3. Results

### 3.1. Correlations and Descriptive Statistics

The results of the correlation analyses, as presented in Table 1, Table 2, Table 3, Table 4 and Table 5, show that PPA has a negative relationship with PAD and a positive relationship with PPEA in all five cohorts. In C1 and C2, the average DPPEA were positive; whereas, the average DPPEA in C3, C4, and C5 were negative. These results indicate that, while the time spent on physical education activity increased from grade 4 in primary school to grade 1 in secondary school, it gradually decreased from grade 2 in secondary school to grade 2 in high school. As no case showed high correlation among all of the measured variables in the five cohorts, the problem of multicollinearity was not an issue. Skewness and kurtosis also met the necessary standards (skewness ±2, kurtosis ±4) [44]; thus, the normality assumption for the measured variables, one of the important assumptions of multiple regression analysis, were met.

### 3.2. Predictions of Perceived Peer Acceptance in Time 2: Main Effects

As seen in Table 6, the input control variable that explains PPA (T2) in Step 1 resulted in an R^2^ value between 0.18 and 0.27 (medium and large effect size [45]). Except for Model 2, SES had a statistically positive effect on PPA (T2). In all of the analysis models, PPEA (T2) had a statistically positive effect on PPA (T2).

In Step 2, the effect of PAD (T1) on PPA (T2) were examined after the effects of the control variables entered in Step 1 were excluded. The results showed that, in all the analysis models, the standardization coefficients β of PAD (T1) that affect PPA (T2) were statistically significant between −0.06 and −0.08. This indicates that a child with high aggressive disposition has a negative effect on peer acceptance one year later, and that this effect was identically present throughout the early teen years.

### 3.3. Predictions of Perceived Peer Acceptance in Time 2: Interactions Effect

The results of the analysis of control effects showed that, in Model 1 during school transition, the slope of a (Gender) × b (PPEA (T2)) × e (PAD (T1)) was statistically significant (β = −0.60, t = 2.50, *p* = 0.000). To understand the three-way interaction effect, PPEA (T2) was divided into a high group with Mean (3.08) + 1 SD (1.14) and low group Mean (3.08)—1 SD (1.14). Figure 1 shows the standardized coefficients of the simple regression of the effect of PAD (T1) on PPA (T2) by gender. Among the male students, the negative relationship of PAD (T1) with PPA (T2) was not greatly affected by PPEA; whereas, higher PPEA was associated with a stronger negative relationship among the female students.

In Model 3, which is during the non-transition phase, the slope of c (DPPEA) × e (PAD (T1)) was statistically significant (β = 0.40, t = 2.10, *p* = 0.036). DPPEA was divided into a high group with Mean (3.08) + 1 SD (1.14) and a low group with Mean (3.08)—1 SD (1.14) to understand the two-way interaction effect, after which the standardized coefficients of a simple regression were calculated. The results showed that, while the standardized coefficient (β) of the effect of PAD (T1) on PPA (T2) was −0.31 in the low DPPEA group, it was −0.16 in the high DPPEA group. Thus, adolescents who engaged in higher physical education activities over a one-year period had weakened negative influence on the effect of aggressive disposition on peer acceptance.

## 4. Discussion

The present study examined the effect that aggression had on peer acceptance based on gender and physical education activity, and whether this effect varied during school transition and non-transition.

### 4.1. Role and Significance of the Control Variable

Table 6 shows that Step 1 of the Hierarchical Multiple Regression Analysis for all models was found to have an R^2^ that ranged from 0.18 to 0.27; this means that the control variable in Step 1 explains the range of 18% to 27% of the total distribution of the dependent variable. In our study, we eliminated the effect the control variable has on the dependent variable by including it in Step 1, after which we sequentially tested the effect aggression had on peer acceptance in Step 2 and the statistical significance of the control variable in Step 3. In other words, the existence of direct effect and moderating effect was individually tested after eliminating the distribution that explains the dependent variable by inputting the control variable, which thereby increases the validity and reliability of the research results by following statistical procedures.

A positive relationship with peer acceptance was noted in all analysis models in Table 6, except Model 2, when inputting SES as a control variable in our study. Adolescents who perceived financial sufficiency maintained good peer relationships, which supports the previous results that SES influences sports participation in adolescents and their social standing in the peer group [42,43].

In accordance with social interdependence theory [25,26], physical education activity had the potential to improve adolescents’ peer acceptance, which was supported by the results of our study. All analysis models in Table 6 showed that physical education activity positively predicts peer acceptance, which indicates that physical education activity helps in the formation of social network in early to late adolescence. Besides, DPPEA and peer acceptance had a negative relationship in Model 1 and Model 4, indicating that peer acceptance is worse when the increase in physical education activity is lower when compared to the previous year. According to research by Shakib, Veliz, Dunbar, and Sabo [46], one of the ways to gain high popularity among peers in childhood and adolescence is being good at sports. Similar to the results that were reported by Daniel and Leaper [27], sports participation in adolescents is shown to have a strong relationship with peer relationship and acceptance, which is in line with the results of our study.

### 4.2. Understanding the Moderating Variables of Gender and Physical Education Activities

In Model 1 in Table 6, the effect of aggressive disposition on peer acceptance was shown to differ during non-transition based on gender and physical education activity. For male students, the negative relationship between PAD (T1) and PPA (T2) remained largely unchanged, regardless of physical education activity; whereas, for female students, the negative relationship between PAD (T1) and PPA (T2) strengthened with increased physical education activity. Thus, increased participation in physical education activities does not seem to improve the negative relationship between aggression and peer acceptance; instead it has a negative influence on the relationship between aggression and peer acceptance for female students.

Although it is unclear why physical education activities of female students negatively impact the relationship between aggression and peer acceptance, it is thought that the perception of sports as an activity that emphasizes masculinity has an impact [47]. During adolescence, male and female students begin to perceive gender-based social norms that dictate that male students must be macho and tough. Gender-role expectations are said to lead to negative response from peers when a behavior that violates social norms is noted [3,48]. While considering that physical education in Korea is centered around sports that emphasize masculinity and easily isolate female students [49,50], female students who participate in physical education face heightened gender-role expectations when compared to the gender-role expectations from their peers. Thus, physical education activity does not strengthen the relationship between aggression and peer acceptance.

Physical education activity of male students should positively impact the negative relationship between aggressive disposition and peer relationship, since physical education is centered on sports, which emphasize masculinity. However, this conclusion was not supported in our study for early adolescence. During early adolescence, peer bullying is used as an avenue to seek influence within peer group [51], and it tends to serve as an avenue to achieve status via aggression among students who have influence in school [52]. In this vein, physical education that emphasizes masculinity in early adolescence can be used as an avenue by male students to seek influence within the peer group. Thus, based on our results, physical education activity could strengthen the negative tendency of aggressive disposition to predict peer acceptance.

### 4.3. Physical Education Activities During School Transition and Non-transition from a Developmental Perspective

While R^2^ in Step 1 of Models 1, 3, and 5 in Table 6 during non-transition was between 0.23 and 0.27, R^2^ in Step 1 of Models 2 and 4 was 0.18; hence, analysis models during non-transition showed a higher R^2^. One of the reasons that R^2^ is generally higher during non-transition than during school transition is that the effect of control variable PPA (T1) on dependent variable PPA (T2) is higher during non-transition. It is predicted that the relationship between PPA (T1) and PPA (T2) is higher during non-transition than during school transition, because, during non-transition, a stable social network is continuously formed among peers, while a new social network must be formed with new peers during school transition.

Despite the fact that R^2^ of Step 1 in Models 2 and 4 during school transition is lower than the R^2^ of Models 1, 3, and 5 during non-transition in Table 6, the standardized coefficients of PPEA (T2) predicting PPA (T2) were higher during school transition than during non-transition. These results allow the inference that adolescents used physical education activity as a useful avenue for the improvement of peer relationship more frequently during school transition than during non-transition. Perron-Gélinas et al. [3] implied that sports participation can be used as an important avenue for improving peer relationships during school transition, which necessitates the renegotiation of social status, and our results empirically support their claim.

The three-way interaction effect of Model 1 (during non-transition) showed that physical education activity in female students negatively impacts the relationship between aggressive disposition and peer acceptance, but this was not observed in Model 2 (during school transition). In other words, physical education activity likely did not help to improve the negative relationship between aggressive disposition and peer acceptance during non-transition between grades 4 and 5 in primary school; instead, it probably strengthens the tendency of aggressive disposition to predict low peer acceptance for female students who participate in more physical education activities. However, this negative impact of physical education activity did not exist between grade 6 in primary school and grade 1 in secondary school (during school transition). Therefore, physical education activity during school transition seems to help improve the negative relationship between aggressive disposition and peer acceptance for female students in early adolescence.

In contrast, the positive impact of physical education activity during school transition was not seen in mid and late adolescence. From grade 2 to grade 3 in secondary school (Model 3, non-transition), an increase in physical education activity in comparison to previous years worsened the negative impact of aggressive disposition on peer acceptance. However, from grade 3 in secondary school to grade 1 in high school (Model 4, school transition), increased physical education activity was not likely to help improve the negative relationship between aggressive disposition and peer acceptance. Further, Model 5 showed similar results. One of the reasons for the disappearance of positive impact physical education activity increase had on the relationship between aggressive disposition and peer acceptance, which existed during non-transition (Model 3) in Model 4 and Model 5, could be the steep decrease in physical education activities.

It was noted that physical education activities decreased the most after grade 3 in secondary school on examining the average score of physical education activities that are provided in Table 1, Table 2, Table 3, Table 4 and Table 5. This can be seen as a result of the reduction in emphasis on physical education after high school in Korea, and an increased emphasis on college admissions examinations. Until grade 3 in secondary school, there was sufficient physical education class time, which allowed the positive influence of physical education activity to be present; in contrast, after grade 3 in secondary school, there was not enough time to perform sufficient physical education activities, and thus physical education activity could not help to improve the negative relationship between aggressive disposition and peer acceptance.

### 4.4. Advantages and Limitations of Our Study and Future Research Directions

Our study took a longitudinal approach for examining the relationship between aggressive disposition and peer acceptance based on gender and physical education activities based on five sets of cohort data, each reflecting school transition and non-transition. In recent years, there have been studies on the relationship between aggression and peer acceptance via a longitudinal approach or developmental perspective [3,27]; however, there has not been a study that differentiated between school transition and non-transition across the early adolescence, as we have done in ours.

On examining the composition of the survey in the panel data used in our study, it was noted that there are many items that measure a variety of areas other than the measurement variables (ex: physical development, intellectual development, socioemotional development, delinquency, living time, family environment, friend relationships, educational environment, community, media environment, activity/culture) and, thus, the possibility of measurement error bias due to the context of items is low. If physical education activity, aggression, and peer acceptance items are placed together, it is possible for respondents to implicitly predict the relationship between the variables and respond; however, with various filter items (example: items that measure different areas) present, the measurement bias due to implicit theory is less likely to be observed. Therefore, the present study was conducted with a longitudinal approach from grade 4 in primary school to grade 2 in high school, including both school transition and non-transition periods, and it has a low measurement error bias due to the various filler items that can have a positive impact on the validity and reliability of the research results.

Many related studies have been conducted regarding aggression and peer acceptance while using peer nomination [3,4,53,54]. However, the aggression and peer acceptance in our study were measured via a self-report survey, which makes social desirability bias possible. Future studies should focus on the effect that physical education activity has on the relationship between aggression and peer acceptance during school transition and non-transition using peer nominations. Finally, our study measured aggression in early teen students, and thus was not able to take a detailed look at the effect that overt or relational aggression forms have on peer acceptance. It is necessary to classify aggression into overt and relational aggression to test whether their effects on peer acceptance differ by gender and physical education activity in the future.

## 5. Conclusions

In conclusion, the results of this study suggest that, in early adolescence, the negative relationship between aggression and peer acceptance is strengthened for female students who participate in more physical education activities when compared to those who do not. The results also indicate that physical education activity does not seem to improve the negative relationship between aggression and peer acceptance for male students. However, during transition from primary to secondary school, we did not find any negative effects of physical education activity; moreover, for middle-adolescents, for whom physical education activity increases in comparison to previous years, the negative relationship between aggression and peer acceptance was weakened. Therefore, based on our results, physical education activity can be considered to be a promising avenue for improving social skills in teenagers.

## Figures and Tables

**Figure 1 ijerph-16-03190-f001:**
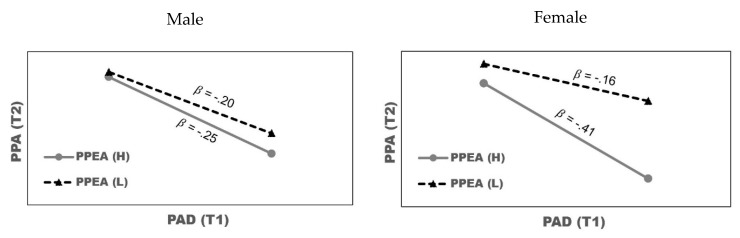
The effect of Perceived Aggressive Disposition (PAD) on Perceived Peer Acceptance (PPA) by gender and participation in physical education activities. Note: PPEA = perceived physical education activity; PAD = perceived aggressive disposition; PPA = perceived peer acceptance; H = high; L = low.

**Table 1 ijerph-16-03190-t001:** Correlation and statistical analyses of observed variables for Cohort 1.

Variables	SES	PPEA (T1)	PPEA (T2)	DPPEA	PPA (T1)	PPA (T2)	PAD (T1)
SES	-						
PPEA (T1)	0.02	-					
PPEA (T2)	0.07 **	0.25 ***	-				
DPPEA	0.04	−0.59 ***	0.64 ***	-			
PPA (T1)	0.12 ***	0.15 ***	0.11 ***	−0.03	-		
PPA (T2)	0.22 ***	0.12 ***	0.13 ***	0.02	0.46 ***	-	
PAD (T1)	−0.11 ***	−0.03	−0.04	−0.01	−0.27 ***	−0.21 ***	-
Mean	5.11	2.83	3.08	0.25	3.27	3.29	1.68
(SD)	(1.14)	(1.09)	(1.14)	(1.37)	(0.61)	(0.56)	(0.63)
Skewness	0.08	0.47	0.17	−0.05	−0.68	−0.65	0.82
Kurtosis	−1.04	−0.66	−0.95	−0.02	0.30	0.50	0.40

Note: SES = Socioeconomic status; PPEA = perceived physical education activity; DPPEA = degree of perceived physical activity; PPA = perceived peer acceptance; PAD = perceived aggressive disposition; ** *p* < 0.01, *** *p* < 0.001.

**Table 2 ijerph-16-03190-t002:** Correlation and statistical analyses of observed variables for Cohort 2.

Variables	SES	PPEA (T1)	PPEA (T2)	DPPEA	PPA (T1)	PPA (T2)	PAD (T1)
SES	-						
PPEA (T1)	−0.01	-					
PPEA (T2)	−0.01	0.29 ***	-				
DPPEA	−0.01	−0.51 ***	0.67 ***	-			
PPA (T1)	0.00	0.19 ***	0.17 ***	0.01	-		
PPA (T2)	−0.03	0.12 ***	0.20 ***	0.09 ***	0.40 ***	-	
PAD (T1)	−0.04 *	−0.05 *	−0.05 *	−0.00	−0.14 ***	−0.13 ***	-
Mean	6.13	3.08	3.33	0.25	3.20	3.17	2.06
(SD)	(0.51)	(1.12)	(1.31)	(1.45)	(0.58)	(0.53)	(0.70)
Skewness	−0.09	0.08	−0.31	−0.16	−0.74	−0.44	0.31
Kurtosis	2.07	−0.87	−1.06	−0.02	1.03	1.23	−0.17

Note: SES = Socioeconomic status; PPEA = perceived physical education activity; DPPEA = degree of perceived physical activity; PPA = perceived peer acceptance; PAD = perceived aggressive disposition; * *p* < 0.05, *** *p* < 0.001.

**Table 3 ijerph-16-03190-t003:** Correlation and statistical analyses of observed variables for Cohort 3.

Variables	SES	PPEA (T1)	PPEA (T2)	DPPEA	PPA (T1)	PPA (T2)	PAD (T1)
SES	-						
PPEA (T1)	0.09 ***	-					
PPEA (T2)	0.09 ***	0.46 ***	-				
DPPEA	0.01	−0.49 ***	0.55 ***	-			
PPA (T1)	0.21 ***	0.21 ***	0.18 ***	−0.02	-		
PPA (T2)	0.15 ***	0.13 ***	0.17 ***	0.05 *	0.51 ***	-	
PAD (T1)	−0.11 ***	0.05 *	−0.05 *	0.00	−0.24 ***	−0.21 ***	-
Mean	4.48	3.29	3.13	−0.16	3.18	3.18	1.94
(SD)	(0.98)	(1.32)	(1.49)	(1.42)	(0.51)	(0.50)	(0.73)
Skewness	0.57	−0.28	−0.12	−0.18	−0.23	−0.23	0.26
Kurtosis	0.43	−1.11	−1.25	0.44	0.57	1.02	−0.29

*Note:* SES = Socioeconomic status; PPEA = perceived physical education activity; DPPEA = degree of perceived physical activity; PPA = perceived peer acceptance; PAD = perceived aggressive disposition; * *p* < 0.05, *** *p* < 0.001.

**Table 4 ijerph-16-03190-t004:** Correlation and statistical analyses of observed variables for Cohort 4.

Variables	SES	PPEA (T1)	PPEA (T2)	DPPEA	PPA (T1)	PPA (T2)	PAD (T1)
SES	-						
PPEA (T1)	0.60 *	-					
PPEA (T2)	0.07 *	0.29 ***	-				
DPPEA	0.00	−0.69 ***	0.50 ***	-			
PPA (T1)	0.11 ***	0.15 ***	0.11 ***	−0.06 *	-		
PPA (T2)	0.17 ***	0.13 ***	0.16 ***	−0.00	0.38 ***	-	
PAD (T1)	−0.12 ***	−0.06 *	−0.03	0.03	−0.09 ***	−0.11 ***	-
Mean	4.06	2.98	2.63	−0.35	3.11	3.12	2.23
(SD)	(0.90)	(1.34)	(1.13)	(1.48)	(0.51)	(0.47)	(0.67)
Skewness	0.28	0.08	0.31	−0.03	−0.40	−0.36	0.09
Kurtosis	1.71	−1.17	−0.51	−0.17	1.02	1.28	−0.21

*Note:* SES = Socioeconomic status; PPEA = perceived physical education activity; DPPEA = degree of perceived physical activity; PPA = perceived peer acceptance; PAD = perceived aggressive disposition; * *p* < 0.05, *** *p* < 0.001.

**Table 5 ijerph-16-03190-t005:** Correlation and statistical analyses of observed variables for Cohort 5.

Variables	SES	PPEA (T1)	PPEA (T2)	DPPEA	PPA (T1)	PPA (T2)	PAD (T1)
SES	-						
2. PPEA(T1)	0.07 **	-					
3. PPEA (T2)	0.08 **	0.38 ***	-				
4. DPPEA	0.00	−0.55 ***	0.57 ***	-			
5. PPA (T1)	0.17 ***	0.15 ***	0.12 ***	−0.02	-		
6. PPA (T2)	0.16 ***	0.12 ***	0.17 ***	0.05 *	0.46 ***	-	
7. PAD (T1)	−0.06 **	−0.02	−0.06 **	−0.03	−0.17 ***	−0.15 ***	-
Mean	4.06	2.64	2.50	−0.15	3.13	3.12	1.97
(SD)	(0.90)	(1.13)	(1.15)	(1.28)	(0.47)	(0.44)	(0.61)
Skewness	0.25	0.31	0.46	0.00	−0.33	−0.13	0.21
Kurtosis	1.70	−0.54	−0.44	0.77	1.11	1.48	−0.36

*Note:* SES = Socioeconomic status; PPEA = perceived physical education activity; DPPEA = degree of perceived physical education activity; PPA = perceived peer acceptance; PAD = perceived aggressive disposition; * *p* < 0.05, ** *p* < 0.01, *** *p* < 0.001.

**Table 6 ijerph-16-03190-t006:** Hierarchical multiple linear regression predicting perceived peer acceptance in Time 2.

Step	Variables	Model 1 (C1)		Model 2 (C2)		Model 3 (C3)		Model 4 (C4)		Model 5 (C5)	
		β	R^2^ (ΔR^2^) [95% CI]	β	R^2^ (ΔR^2^) [95% CI]	B	R^2^ (ΔR^2^) [95% CI]	B	R^2^ (ΔR^2^) [95% CI]	B	R^2^ (ΔR^2^) [95% CI]
1	SES	0.16 ***	0.25 *** [0.21, 0.28]	−0.03	0.18 *** [0.15, 0.21]	0.05 *	0.27 *** [0.23, 0.30]	0.13 ***	0.18 *** [0.14, 0.21]	0.07 ***	0.23 *** [0.20, 0.26]
	Gender^a^	0.04		0.03		0.04		0.04		0.06 **	
	PPEA (T2)^b^	0.11 ***		0.16 ***		0.09 **		0.15 ***		0.15 ***	
	DPPEA^c^	−0.05 *		−0.02		0.01		−0.06 *		−0.03	
	PPA (T1)^d^	0.43 ***		0.38 ***		0.48 ***		0.34 ***		0.43 ***	
2	PAD (T1)^e^	−0.08 ***	0.25 (0.006 ***) [0.22, 0.29]	−0.07 **	0.19 *** (0.005 **) [0.16, 0.22]	−0.09 ***	0.27 *** (0.007 ***) [0.24, 0.31]	−0.06 ***	0.18 *** (0.003 *) [0.15, 0.21]	−0.07 **	0.24 *** (0.004 **) [0.20, 0.27]
3	a × b	0.32	0.26 *** (0.007 **) [0.23, 0.30]	−0.31	0.19 *** (0.002) [0.16, 0.22]	0.08	0.28 *** (0.003) [0.24, 0.31]	0.01	0.18 *** (0.002) [0.15, 0.21]	−0.06	0.24 *** (0.003) [0.21, 0.27]
	a × e	0.31		−0.26		0.30		0.18		0.19	
	b × e	0.37		−0.37		0.33		0.18		0.50	
	a × b × e	−0.60 ***		0.24		−0.23		−0.17		−0.53	
	a × c	0.27	0.25 *** (0.003) [0.22, 0.29]	−0.33	0.19 *** (0.003) [0.16, 0.22]	0.24	0.28 *** (0.003) [0.24, 0.31]	−0.11	0.18 *** (0.001) [0.15, 0.21]	−0.14	0.24 *** (0.003) [0.21, 0.27]
	a × e	−0.13		−0.04		0.07		0.06		−0.06	
	c × e	0.29		−0.38		0.40 *		−0.16		−0.04	
	a × c × e	−0.32		0.24		−0.27		0.19		−0.00	

*Note:* SES = Socioeconomic status; PPEA = perceived physical education activity; DPPEA = degree of perceived physical activity; PPA = perceived peer acceptance; PAD = perceived aggressive disposition; Gender (male = 0; female = 1); CI = confidence interval. * *p* < 0.05, ** *p* < 0.01, *** *p* < 0.001.

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
