# Peer review of "Effect of Aggression on Peer Acceptance Among Adolescents During School Transition and Non-Transition: Focusing on the Moderating Effects of Gender and Physical Education Activities"

_ijerph, 2019, doi:10.3390/ijerph16173190_

Round 1

Reviewer 1 Report

I have had a look at the changes, and see that authors have attempted to moderate their findings given that they did not use an experimental design, which was my main concern. Therefore authors' claims are now more reasonable. However their way round this caveat has been to use complicated phrases which are hard to follow, and unnecessary : there are several uses of "did not positively influence the negative....."

From a content point of view, I can accept this article, but I suggest the editorial team work with authors to simplify the language to make it more readable.  For example, they can simply remove the 'positively from these sentences, or replace 'positively influence' with 'improve'.

Author Response

Thank you for the opportunity to improve our manuscript.

We have seriously considered each of the comments provided by Reviewer 1

Red text in the revised manuscript denotes text referred to in a response to reviewer feedback.

Comments #1: I have had a look at the changes, and see that authors have attempted to moderate their findings given that they did not use an experimental design, which was my main concern. Therefore authors' claims are now more reasonable. However their way round this caveat has been to use complicated phrases which are hard to follow, and unnecessary : there are several uses of "did not positively influence the negative....."

From a content point of view, I can accept this article, but I suggest the editorial team work with authors to simplify the language to make it more readable.  For example, they can simply remove the 'positively from these sentences, or replace 'positively influence' with 'improve'.

Based on your comments, we have revised it as follows. Line 307: to improve Line 322 - 324: Thus, increased participation in physical education activities does not seem to improve the negative relationship between aggression and peer acceptance; Line 365 - 368: In other words, physical education activity likely did not help to improve the negative relationship between aggressive disposition and peer acceptance during non-transition between grades 4 and 5 in primary school; Line 371 - 373: Therefore, for female students in early adolescence, physical education activity during school transition seems to help improve the negative relationship between aggressive disposition and peer acceptance. Line 378-380: increased physical education activity was not likely to help improve the negative relationship between aggressive disposition and peer acceptance. Line 389 - 391: there was not enough time to perform sufficient physical education activities, and thus physical education activity could not help to improve the negative relationship between aggressive disposition and peer acceptance. Line 424 - 426: The results also indicate that physical education activity does not seem to improve the negative relationship between aggression and peer acceptance for male students.

Reviewer 2 Report

This present aims to examine the effect of aggression on per acceptance among adolescents. In particular, the authors focused on the moderating effects of gender and participation in physical education activities and examined whether these effects varied during school transition. Overall, the results showed that the interaction effect of gender and physical education activity influenced the relationship between aggression and peer acceptance in only early adolescence. The authors suggested that physical education activities improve the negative relationship between and peer acceptance during school transition. The study is well-designed and well-conducted, and the results are interesting and substantial for the community. However, I have two short questions:

Methods

- To measure the following parameters: PPEA, DPPEA and PPA the authors used self-reported survey. I would like to know if the authors designed the surveys themselves. If so, how did they verify the psychometric characteristics. If not, the authors could provide references to the surveys.

- In the statistical processing section, the authors indicated that they performed a hierarchical multiple regression. I would like them to specify whether they have checked multicollinearity, homoscedasticity, normal residue distribution and error independence.

Author Response

Thank you for the opportunity to improve our manuscript.

We have seriously considered each of the comments provided by Reviewer 2.

The text highlighted in yellow in the revised manuscript denotes text referred to in a response to reviewer feedback.

Comments #1: To measure the following parameters: PPEA, DPPEA and PPA the authors used self-reported survey. I would like to know if the authors designed the surveys themselves. If so, how did they verify the psychometric characteristics. If not, the authors could provide references to the surveys.

We added a reference to the measurement questionnaire. Line 184, 201 Lee, S.; Shin, M.; Smith, A. L. The relationship of physical activity from physical education with perceived peer acceptance across childhood and adolescence.  Sch. Health. 201989, 452-459;DOI:10.1111/josh.12756. Line 191 Choi, B.; Park, S.; Shin, J. The effect of aggression and self-blame style on the longitudinal relationship between perceived peer acceptances: a moderated mediation analysis. Asian J. Educ.2016, 17, 171-194; DOI: 10.15753/aje.2016.09.17.3.171.

Comments #2. In the statistical processing section, the authors indicated that they performed a hierarchical multiple regression. I would like them to specify whether they have checked multicollinearity, homoscedasticity, normal residue distribution and error independence.

Reflecting your comments, the following related contents were added. Line 218 - 222 The analyzed data were found to meet four assumptions of multiple regression analysis: 1. linearity of the phenomenon measured, 2. constant variance of the error terms, 3. independence of the error terms, and 4. normality of the error term distribution. As shown in Tables 1-5, there is no high correlation between variables, so the multicollinearity problem does not exist.

This manuscript is a resubmission of an earlier submission. The following is a list of the peer review reports and author responses from that submission.

Round 1

Reviewer 1 Report

This is an interesting and well-written study. The introduction and rationale is clear, with an interesting hypothesis presented.

The tables were difficult to read, possibly because the variable names weren't in the first row as well as the first column. Using only numbers made it difficult to see which of the seven variable is which. It would be better to have the means etc of all cohorts presented separately, rather than included above and below the correlations.

When comparing R-squared values between models and time points, it would be beneficial to use 95% confidence intervals for R-squared values. For details, see Cohen et al. (2003) Applied multiple regression/correlation analysis for the behavioral sciences, Lawrence Erlbaum Associates, London, UK. This would enable comparisons to be made clearly between models, with statistical differences able to be reported.

It would be worthwhile discussing the magnitude of the effects observed. Are the reported R-squared values small, moderate, or large?

Author Response

Authors’ general response to the Reviewer 1:

Thank you for the opportunity to improve our manuscript.

We have seriously considered each of the comments provided by Reviewer 1 and Reviewer 2. Major modifications are as follows:

The table was revised to improve readability and 95% confidence intervals for R-squared values were also presented (Reviewer 1 suggested). The conclusion section was added (Editor suggested), and the sentence was modified to conjecture, not to the meaning of conviction (Reviewer 2 suggested).

Blue text in the revised manuscript denotes text referred to in a response to reviewer feedback.

The tables were difficult to read, possibly because the variable names weren't in the first row as well as the first column. Using only numbers made it difficult to see which of the seven variable is which. It would be better to have the means etc of all cohorts presented separately, rather than included above and below the correlations.

Author's responses: We have reorganized the tables to make them more readable. Table numbers in the text have also been revised. Line 230 ~ 247

When comparing R-squared values between models and time points, it would be beneficial to use 95% confidence intervals for R-squared values. For details, see Cohen et al. (2003) Applied multiple regression/correlation analysis for the behavioral sciences, Lawrence Erlbaum Associates, London, UK. This would enable comparisons to be made clearly between models, with statistical differences able to be reported.

Authors' responses: We have added a 95% confidence interval for R-squared values in Table 6. Line 255

It would be worthwhile discussing the magnitude of the effects observed. Are the reported R-squared values small, moderate, or large?

Authors' responses: The effect sizes for the R-squared values are presented based on the references.

Line 251 ~ 252

As seen in Table 6, the input control variable that explains PPA (T2) in Step 1 resulted in an R2 value between .18 and .27 (medium and large effect size [43]).

Cohen, J. Statistical Power Analysis for the Behavioral Sciences, 2nd ed.; Lawrence Erlbaum Associates: Hillsdale, NJ, USA, 1988.

Reviewer 2 Report

This is a complex study using secondary data asking several questions and coming to several conclusions. I think the study might have been strengthened by asking less questions.

 Since the study is based on correlations and analysis of longitudinal data, not on experimental research, you cannot conclude that "physical education activities can help improve the negative  relationship between aggression and peer acceptance during school transition". 

Your conclusions need to be less definitely stated, as do some of your conjectures around the anomilies in your results. Further thought on other interpretations of the data might strengthen the discussion.

However overall this is a really meaty paper, and the complex ideas clearly described. I particularly appreciated the background section.

Author Response

Authors’ general response to the Reviewer 2:

Thank you for the opportunity to improve our manuscript.

We have seriously considered each of the comments provided by Reviewer 1 and Reviewer 2. Major modifications are as follows:

The table was revised to improve readability and 95% confidence intervals for R-squared values were also presented (Reviewer 1 suggested). The conclusion section was added (Editor suggested), and the sentence was modified to conjecture, not to the meaning of conviction (Reviewer 2 suggested).

Blue text in the revised manuscript denotes text referred to in a response to reviewer feedback.

Since the study is based on correlations and analysis of longitudinal data, not on experimental research, you cannot conclude that "physical education activities can help improve the negative relationship between aggression and peer acceptance during school transition".

Authors' responses: Reflecting your comments, we revised the sentence to conjecture rather than to the meaning of conviction. Line 27 ~ 28 Thus, this study suggests that physical education activities improve the negative relationship between aggression and peer acceptance during school transition.

Your conclusions need to be less definitely stated, as do some of your conjectures around the anomilies in your results. Further thought on other interpretations of the data might strengthen the discussion.

Authors' responses: The conclusion section is added as follows. we also revised the sentence to conjecture rather than to the meaning of conviction.

Line 417 ~ 426

In conclusion, the results of this study suggest that, in early adolescence, the negative relationship between aggression and peer acceptance is strengthened for female students who participate in more physical education activities compared to those who do not. The results also indicate that physical education activity does not positively influence the negative relationship between aggression and peer acceptance for male students. However, during transition from primary to secondary school, we did not find any negative effects of physical education activity; moreover, for middle-adolescents, for whom physical education activity increases in comparison to previous years, the negative relationship between aggression and peer acceptance was weakened. Therefore, based on our results, physical education activity can be considered a promising avenue for improving social skills in teenagers.

Line 318 ~ 319

Thus, participation in more physical education activities does not seem to positively influence the relationship between aggression and peer acceptance;

Line 338 ~ 339

Thus, based on our results, physical education activity could strengthen the negative tendency of aggressive disposition to predict peer acceptance.

Line 361 ~ 364

physical education activity likely did not positively influence the negative relationship between aggressive disposition and peer acceptance during non-transition between grade 4 and 5 in primary school; instead, it probably strengthens the tendency of aggressive disposition to predict low peer acceptance for female students who participate in more physical education activities.

Line 373 ~ 375

increase in physical education activity was not likely to positively influence the negative relationship between aggressive disposition and peer acceptance.